# Industry 4.0 Roadmap: Implementation for Small and Medium-Sized Enterprises

**Alberto Cotrino \*** , **Miguel A. Sebastián** and **Cristina González-Gaya**

Department of Construction and Manufacturing Engineering, Universidad Nacional de Educación a Distancia (UNED), C/Juan del Rosal 12, 28040 Madrid, Spain; msebastian@ind.uned.es (M.A.S.); cggaya@ind.uned.es (C.G.-G.)

**\*** Correspondence: acotrino3@alumno.uned.es; Tel.: +49-175-815-6948

**Abstract:** The Industry 4.0 era has resulted in several opportunities and challenges for the manufacturing industry and for small and medium-sized enterprises (SME); technologies such as the Internet of Things (IoT), Virtual Reality (VR) or Cloud Computing are changing business structures in profound ways. A literature review shows that most large-sized enterprises have rolled out investment plans, some of which are reviewed during this research and show that Industry 4.0 investments in such companies exceed the turnover of SMEs in all cases (<€50 million), which makes access to those technologies by SMEs very difficult. The research has also identified two gaps: firstly, the recent literature review fails to address the implementation of Industry 4.0 technologies in SMEs from a practical viewpoint; secondly, the few existing roadmaps for the implementation of Industry 4.0 lack a focus on SMEs. Furthermore, SMEs do not have the resources to select suitable technologies or create the right strategy, and they do not have the means to be fully supported by consultancies. To this end, a simple six-step roadmap is proposed that includes real implementations of Industry 4.0 in SMEs. Our results show that implementing Industry 4.0 solutions following the proposed roadmap helps SMEs to select appropriate technologies. In addition, the practical examples shown across this work demonstrate that SMEs can access several Industry 4.0 technologies with low-cost investments.

**Keywords:** Industry 4.0; SME; roadmap; low-cost

## 1. Introduction

Small and medium-sized enterprises (SMEs) have been facing tremendous challenges since 2011 due to the Fourth Industrial Revolution in the form of emerging Industry 4.0 technologies and their applications for the industrial sector [1–3].

Professor Klaus Schwab, Founder and Executive Chairman of the World Economic Forum, coined the phrase "Fourth Industrial Revolution" in 2015 in an article published by Foreign Affairs [4]; it was also the subject of his book published in 2017, in which Schwab highlights a grave concern: that organizations might be unable to adapt [5].

A literature review in recent years shows that SMEs have to be considered separately from large enterprises with regard to Industry 4.0 implementation, because they are less capable of coping with financial, technological and staffing challenges than large companies [6–8]. This is something that has been identified not only in research works, but also in government initiatives, such as "Industrie 4.0" in Germany, which have highlighted this issue [9].

This has resulted in technology-transfer projects specifically for SMEs to pass on Industry 4.0 solutions and in extensive research works concerning SMEs and Industry 4.0, especially in the first two of the three following areas:

- The identification of key Industry 4.0 technologies, challenges and the associated benefits for SMEs (1);
- The maturity level of implementation or classification based on the level of Industry 4.0 implementation in SMEs (2);
- A process model or roadmap for the implementation of Industry 4.0 technologies (3).

Most of these research works aim to investigate SMEs from a theoretical point of view. There is, however, a gap in the practical viewpoint, which is greatly needed for business-oriented implementations, as shown in Table 1. This study aims to develop the research in this field by researching the most efficient way to transform old-fashioned manufacturing from a practical point of view.

**Table 1.** Literature review to identify the research gaps in this area. SMEs: small and medium-sized enterprises.

| Title | Area | Focus on SMEs | Type of Paper | Reference |
|---|---|---|---|---|
| Industry 4.0: Adoption challenges and benefits for SMEs | 1 | Yes | Survey/statistical analysis | [10] |
| Industry 4.0 technology implementation in SMEs—A survey in the Danish-German border region | 1 | Yes | Survey/statistical analysis | [11] |
| Three Stage Maturity Model in SMEs towards Industry 4.0 | 2 | Yes | Conceptual | [12] |
| The future of manufacturing industry: a strategic roadmap toward Industry 4.0 | 3 | No | Conceptual | [13] |
| A Maturity Level-Based Assessment Tool to Enhance the Implementation of Industry 4.0 in Small and Medium-Sized Enterprises | 2 | Yes | Conceptual | [14] |
| Classification of Small- and Medium-Sized Enterprises Based on the Level of Industry 4.0 Implementation | 4 | Yes | Conceptual | [15] |
| Smart Factory of Industry 4.0: Key Technologies, Application Case, and Challenges | 1 | No | Literature review/conceptual | [16] |
| Process model for the successful implementation and demonstration of SME-based industry 4.0 showcases in global production networks | 3 | Yes | Conceptual | [17] |
| A critical review of smart manufacturing & Industry 4.0 maturity models: Implications for small and medium-sized enterprises (SMEs) | 2 | Yes | Literature review | [18] |
| Literature Search of Key Factors for the Development of Generic and Specific Maturity Models for Industry 4.0 | 2 | Yes | Literature review | [19] |
| Challenges of Industry 4.0 Technology Adoption for SMEs: The Case of Japan | 1 | Yes | Conceptual | [20] |
| Problems with the Implementation of Industry 4.0 in Enterprises from the SME Sector | 1 | Yes | Conceptual/survey | [3] |
| Roadmap Industry 4.0—Implementation Guideline for Enterprises | 3 | No | Conceptual | [21] |

Additionally, the few roadmaps described for the implementation of Industry 4.0 lack a focus on SMEs, which leads to a gap in this area, as shown in Table 1.

Access to Industry 4.0 technologies must be simplified for SMEs, especially for microenterprises. SMEs cannot afford to lose money, and they therefore need to choose how to invest it very carefully. SMEs need to hit targets when choosing technologies for the transformation to Industry 4.0, and this research focuses on developing a roadmap to implement Industry 4.0 components in an assembly line, describing some practical examples for each technology, allowing an outdated manufacturing line to be turned into a Smart Assembly Line (SAL).

## 2. Methodology

In this work, we applied a research approach in three stages:

- Stage 1—Literature review conducted in 2019 and the start of 2020 to achieve the following, Section 2.1:

  o To identify the situation of SMEs in Europe;
  o To benchmark the transformation towards Industry 4.0 in large enterprises;
  o To understand Industry 4.0 concepts and technologies and estimate their costs.

- Stage 2—Development of the roadmap for SMEs in 2019, shaping the concept of the Smart Assembly Line (SAL), Sections 2.2 and 2.3.
- Stage 3—Testing and validation of the roadmap by using three different Industry 4.0 technologies in 2020, Section 3.

*2.1. Literature Review*

2.1.1. SMEs in Europe

The European Commission defines three categories of SMEs [22] as stated in Table 2:

**Table 2.** Commission recommendation concerning the definition of SMEs.

| Enterprise Category | Employees | Turnover |
|---|---|---|
| Micro SME | 0 to <10 | <€2 million |
| Small SME | 10 to <50 | <€10 million |
| Medium-sized SME | 50 to <250 | <€50 million |

In 2018, just over 25 million enterprises existed in the EU-28, of which 93% were micro SMEs and 5.9% were small SMEs. SMEs generated 56.4% of the total value-added and 66.6% of employment [23].

These figures show the importance of SMEs, especially micro and small SMEs, in the European economy and its development.

2.1.2. Industry 4.0 Technologies

The analysis of different Industry 4.0 technologies was carried out by reviewing some European guidelines; for instance, those proposed by the Mechanical Engineering Industry Association (VDMA) and the compilation of practical examples created by this association since 2019 [24,25]. Additionally, benchmarking was performed through Internet research to obtain a preliminary view of the top Industry 4.0 performers. Best practices have been identified in companies such as Volkswagen, Amazon, Siemens, etc. [26–29].

There is extensive literature about technologies related to Industry 4.0 [7,9,16,24,30]. Industry 4.0 can be implemented in every industrial sector, from shipbuilding to aerospace and medical devices. Transformation to Industry 4.0, however, means different things for each company in these sectors, and requires different approaches to be achieved. At the same time, every company needs to understand the different components and tools offered in order to determine the correct business strategy [31].

Considering the large number of technologies related to Industry 4.0, the authors have selected four of these technologies to perform a deep dive into the literature review: Cloud Computing, the IoT, Digital Transformation (DT) and VR.

Some authors consider the IoT and Cloud Computing to be the key technologies of Industry 4.0 [16]; others cannot imagine a transition to the Fourth Industrial Revolution without considering several Industry 4.0 pillars at the same time [32]. Several authors highlight the advantages that Cloud Computing offers; for instance, it saves costs and delegates liabilities to third-party providers [33].

Moreover, the development of the IoT is a crucial moment in the history of humanity because it is changing our mindset, culture and the way we live or manufacture. As with the Internet age, there will be a pre-IoT world and a post-IoT world. The IoT era will not be an instantaneous transition, but a gradual and continuous shift during which evolution will never stop [34]. Sensor technologies, which are a key element of the IoT, have experienced rapid development in recent times and still face some challenges, such as proper standardization [35]. This development has been driven by the advent of high-speed and low-cost electronic circuits, a change in the way we approach signal processing and corresponding advances in manufacturing technologies [36].

Additionally, there are some research works and conference proceedings that have stated the belief that VR will greatly support the accessibility of smart manufacturing [37–39].

Based on the abovementioned literature review, the authors came to the following conclusions:

- The IoT has experienced a tremendous cost reduction in the past few years, and the offered technologies have multiplied.
- DT is very generic and includes too many technologies and components; thus, it is not suitable for further analysis in this paper.
- Cloud Computing is now a well-established technology that is offered at a low price.
- VR is a very flexible technology that can be implemented in several areas of production.

Figure 1 shows the interconnection between the different components of these four technologies, with some branches (inner circle) replicated in some stems/leaves (outer circle). This is a positive aspect because the implementation of one technology will provide access to other technologies.

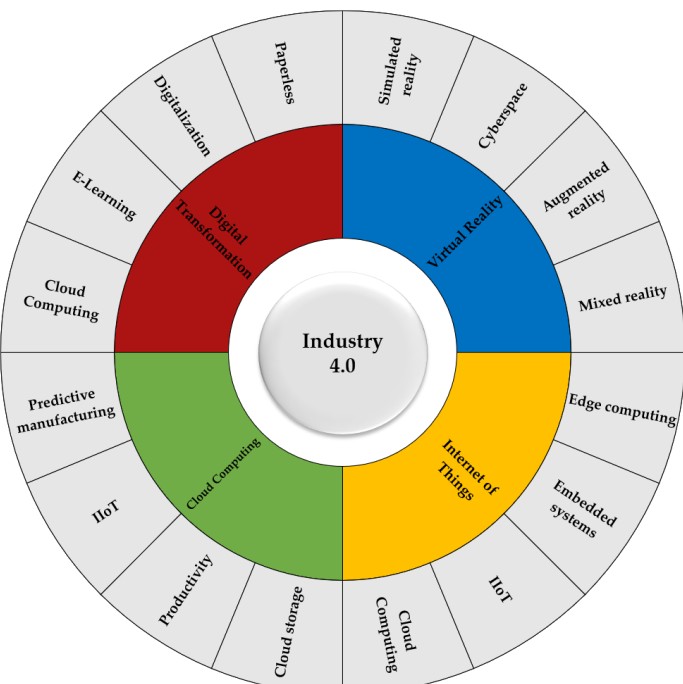

**Figure 1.** Sunburst and interconnection of Industry 4.0 components. IIoT: Industrial Internet of Things.

The outlook for the coming years, according to data extracted from the European Patent Office [40] in Figure 2, shows that changes are still ongoing. The numbers of patents in the selected technologies (Cloud Computing, IoT and VR) has greatly increased over the past seven years. This will lead to cost reductions and will enable access to these technologies by more companies.

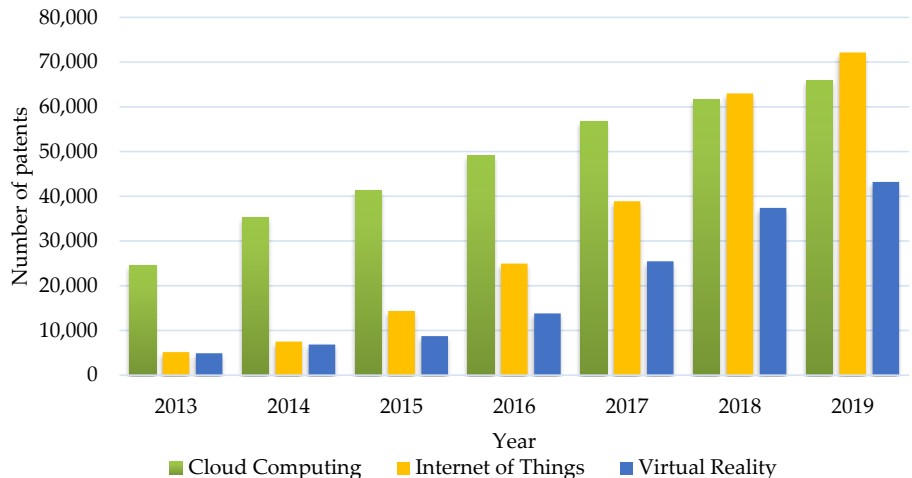

**Figure 2.** Number of patents by Industry 4.0 components.

### 2.1.3. Private Investment in Industry 4.0—Large-Sized Companies

Large-sized companies in Europe have rolled out investment plans and roadmaps for DT and IoT that will last until 2022–2023.

These companies are willing to digitalize business processes, implement agile working methods, help reduce the burden on employees, improve productivity and speed up their processes. Tasks that used to be performed manually will be simplified through improved IT [26]. Personal relationships between the company and the customer are to be made online, and this development will lead to the development of new professions that do not currently exist [41].

As shown in Figure 3, no matter the business sector (telecommunications, banking, automobile industry, food industry, technology, medical devices, insurance or clothing, etc.), every large-sized companies have a roadmap for digital transformation and the implementation of Industry 4.0 technologies, and they are in the process of digitizing the supply chain, customer services and finances.

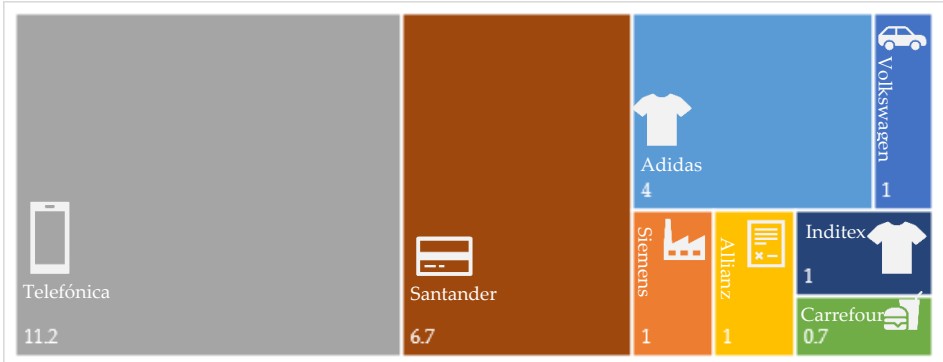

**Figure 3.** Examples of private investment in Industry 4.0 (annually—€ billion).

### 2.1.4. Private Investment in Industry 4.0—SMEs

SMEs do not have the economic resources to implement Industry 4.0 technologies. For this study, the authors targeted microenterprises and small enterprises, based on the definitions given in Table 2, with the selected turnover range being between €0.5 million and €10 million.

The average profit margin of SMEs in 2019 was 7.4%. Of this profit, the average re-investment in innovations was 10% [42]. The assumption is that SMEs are willing to re-invest between €3700 and €74,000 in innovations. The cost of the tools used for this study, based on the three technologies, fall within this range and, at best, must be less than €14,800 to cover the complete range of microenterprises which have a turnover between €0.5 million and €2 million.

### 2.1.5. COVID-19, Key Technologies Supporting Remote Working

The global COVID-19 pandemic has shown the importance of companies embracing remote working [43]. The three technologies selected during this research support a remote working environment:

- In order to enable remote working, the company's information and resources must be available in integrated management software, and companies must embrace DT in the form of Cloud Computing.
- The IoT enables machines and working cells installed in the production line of SMEs to send data to the gateways for further data analysis at the edge or on the Cloud. This not only provides real-time visibility for the operations but also enables the remote monitoring of production.
- VR solutions empower companies to perform activities such as training or prototype verification in a remote environment.

### 2.2. Roadmap

Based on a systematic review, the benefits of Industry 4.0 have been well identified in Sections 1 and 2.1.2. There is, however, an important gap as regards the strategies required to deploy these technologies within SMEs, as mentioned in Section 1. All research works addressing these roadmaps make no distinction between large companies and SMEs and consider a large company structure and expertise for these roadmaps, which ends up involving several departments and teams, such as strategic management, procurement, human resources, Industry 4.0 transition teams or production. SMEs, however, do not have these kinds of structures; they are single-layer companies. Microenterprises, for instance, have fewer than ten employees, and every single employee—rather than teams—should be able to use the roadmap. Many different approaches to roadmapping have been developed, and roadmaps can take many shapes and forms. Generally, the roadmap focuses on a graphic representation that provides a strategic overview of the topic at hand. The authors have selected the templates and guidelines proposed by the Institute for Manufacturing (IfM) of the University of Cambridge [44–46] for the basic development of this roadmap. This holistic roadmap framework links directly to the following fundamental questions that apply in any strategic context:

- Where do we want to go? Where are we now? How do we get there?
- Why do we need to act? What should we do? How should we do it? By when?

These questions are addressed during the first two steps of the six-step roadmap. Additionally, as the research wants to address the practical viewpoint, steps 2–5 present a business-oriented implementation strategy that takes into consideration the limitations of SMEs (e.g., budget, personnel, etc.), as shown in Sections 1 and 2.1.2.

Step 0—Identify bottlenecks

The roadmap begins by identifying those areas of production that are hindering the overall efficiency. The search for bottlenecks consists of evaluating Key Performance Indicators (KPIs) throughout production. This evaluation will reveal the answers to the following questions: "Why do we need to act?" and "Where are we now?".

Step 1—Develop a strategy

Step 1 is a crucial stage for the proper development of a roadmap. During this step, the person in charge at the SME must propose possible long-term countermeasures for these bottlenecks using selected Industry 4.0 technologies (Cloud Computing, IoT or VR) that best suit this purpose. This person must also consider the maximum available budget for the SME when creating this strategy, as already stated in Section 2.1.4. This step provides an answer to the following questions: "Where do we want to go?", "How do we get there?", "What should we do?" and "By when?".

Step 2—Ideas and prototypes

The roadmap continues with the initial deployment of the tool selected in the production line. This initial roll-out begins with a prototype, not by installing the tool across the entire production line. The success of the prototype must be evaluated during a defined time span. If the prototype does not show promising results, the process must return to step 1 in order to propose different long-term countermeasures.

Step 3—Connect/plug-in devices

If the results during step 2 are satisfying, the process continues by deploying the solution across the entire production line. This phase starts by training the operators in the production line on how the tool is used and carrying out some dry runs. This step will improve the acceptance of the selected Industry 4.0 technology by the operators.

Step 4—Analyze

The original KPIs identified during step 1, and how they are measured, may no longer be adequate. During this process, additional KPIs must be proposed and Cloud Computing must be ready to support the storage and analysis of data that the technology selected will generate.

Step 5—Go live

The final step is the official roll-out of the technology across the entire production line. This step focuses on monitoring the defined KPIs. The person in charge at the SME will control sustainment during a defined period.

In short, access to Industry 4.0 technologies must be simplified for SMEs, and the proposed six-step roadmap shown in Figure 4 therefore aims to facilitate decision-making and access to Industry 4.0 technologies in the production area of SMEs. This roadmap has been verified in several examples of practical implementation in the supply chain of SMEs. The results are documented in the following Section 3.

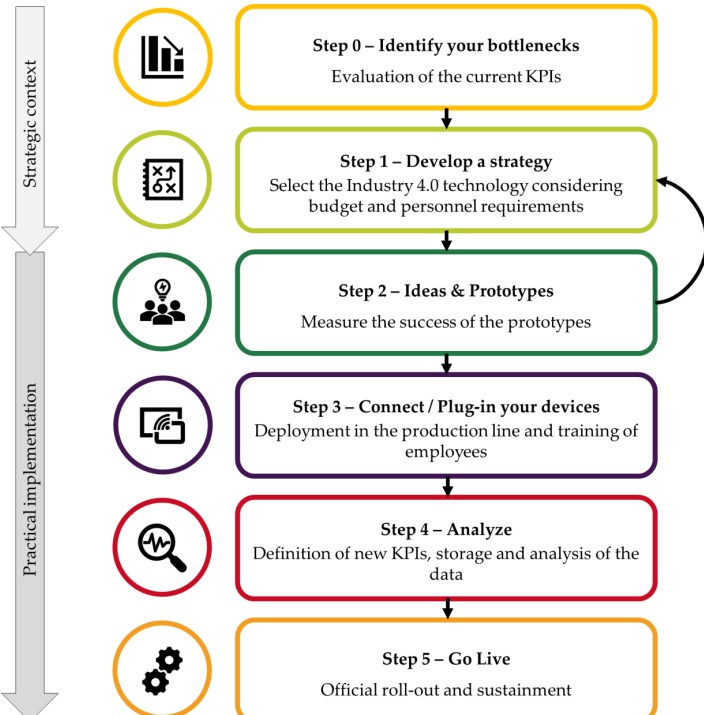

**Figure 4.** Roadmap for the Fourth Industrial Revolution for SMEs. KPIs: Key Performance Indicators.

### 2.3. Smart Assembly Line (SAL)

The SAL is a term that is shaped during this research and further develops the Smart Production Line concept presented in a conference paper in 2017 [47]. The SAL aims to connect the theoretical perspective of Industry 4.0 with the three technologies (Cloud Computing, IoT and VR) selected in this paper so that these technologies can be implemented not in isolation but as a group. The three technologies are capable of fostering the four key characteristics of the SAL, as shown in Figure 5 and their implementation should share a common end goal: that these technologies and solutions should optimize production, enhance productivity and efficiency and be implemented using the theoretical framework of the roadmap defined in Section 2.2.

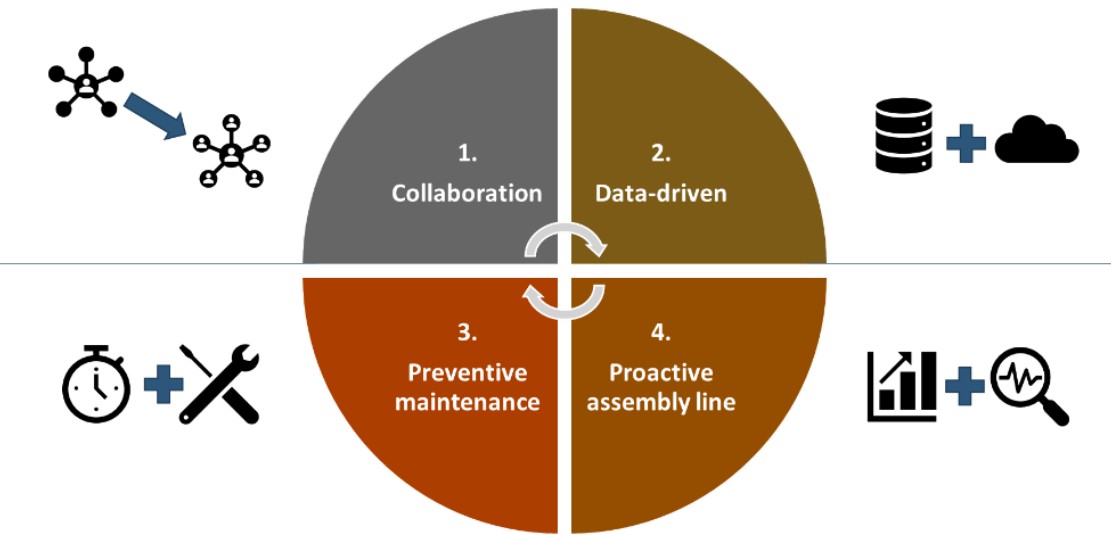

**Figure 5.** Key characteristics of the Smart Assembly Line.

The SAL and the roadmap work together: every component of the technology will be selected and implemented using the roadmap, and once the component is implemented in production, it will foster the characteristics promoted by the SAL, as shown in Figures 5 and 6.

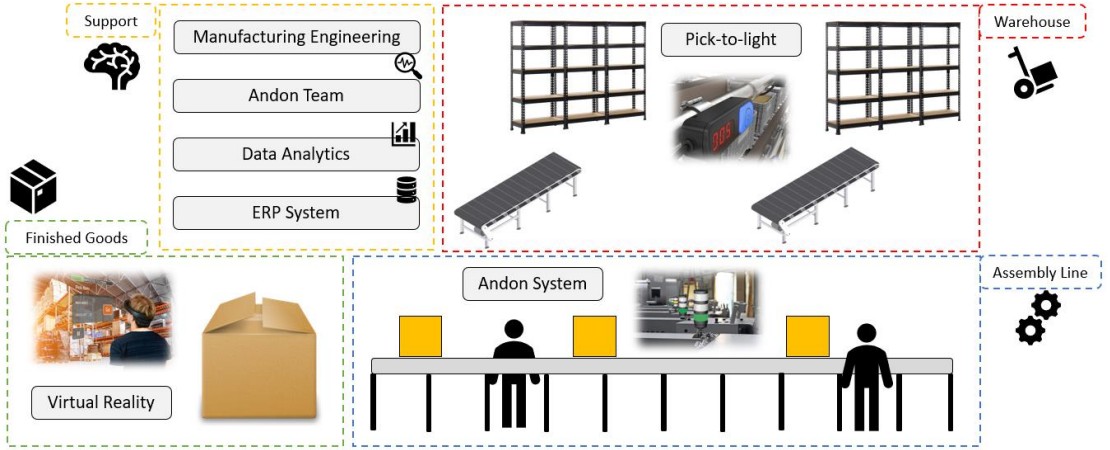

**Figure 6.** Overview of the Smart Assembly Line.

The SAL has four key characteristics that set it apart from a traditional production line.

1. Collaboration: It promotes the integration of all equipment and personnel working in the assembly line. It transforms an assembly line in which all components are working in isolation into an interconnected assembly line.

2.  Data-driven: The SAL generates real-time data and stores them in the Cloud (manufacturing time, first pass yield, number of defective parts, etc.). This data can be retrieved at any time, from anywhere.

3.  Preventive maintenance: Data usage enables the implementation of preventive maintenance in order to avoid down-time.

4.  Proactive assembly line: The SAL identifies the efficiency of the resources and equipment used in the line in real time. If the assembly line is not working efficiently, the line itself will propose a new distribution of the resources to increase productivity.

## 3. Results

### 3.1. Testing and Verification

The study of the Industry 4.0 solutions selected in this research, including testing and verification, was carried out at the Munich manufacturing site of a biomedical company. The Andon system in Section 3.2 has been completely rolled out in the production line, and the results of the validation are conclusive. The mixed reality in Section 3.3 and pick-to-light in Section 3.4 technologies are still at the analysis stage of the six-step roadmap; therefore, no quantitative results are yet available, and only qualitative results of the first prototype will be presented.

### 3.2. Andon System—The IoT Applied to Quality Control

Quality is the main driver of an exceptional production plant. Non-conformances must be kept under control in order to provide customers with the best quality and to maintain favorable productivity.

Real-time visibility and immediate intervention regarding waste are necessary to manage non-conformances. Furthermore, tracking trends in performance over time is also crucial to improve the efficiency of the operations in the assembly line.

Andon is a term that refers to a system that notifies users of a quality or processing problem. An Andon system is one of the elements described in the Jidoka quality control method pioneered by Toyota as part of the Toyota Production System and belongs to the lean production approach [48].

The Andon system proposed within this paper uses a TL70 Modular Tower Light with a wireless radio base connected to a DXM700 Wireless Controller. The data are pushed to the Banner Cloud Data Services (CDS) to create a dashboard and perform Cloud Computing activities.

The Andon system uses two of the technologies selected in Section 2.1.2: the IoT and Cloud Computing. Implementation following the roadmap is described in Table 3.

**Table 3.** Andon system—roadmap application.

| Step Number | Step Description | Application for This Solution |
| --- | --- | --- |
| 0 | Identify bottlenecks | The number of quality non-conformances has been increasing for a long time. Quality failures shall be detected immediately, and countermeasures shall be applied as soon as possible. |
| 1 | Develop a strategy | Every work cell in the assembly line will get a traffic light that is manually switched to notify users of quality non-conformances. It will generate real-time data and will allow employees to resolve the issues in a short period of time. |
| 2 | Ideas and Prototypes | "Think big but start small": A model work cell was created in which the first prototype was tested. |
| 3 | Connect/plug-in your devices | A traffic light was installed and connected to the controller, the dashboard was set and the Andon team was ready to analyze the data. Some dry runs were performed until the system ran stably. |
| 4 | Analyze | Andon team created daily KPIs to control the evolution of quality non-conformances and analyze the results, asking the question "Am I winning or am I losing?". |
| 5 | Go live | Once the results were verified and the Andon system proved that the IoT could help to track and reduce quality non-conformances, the Andon system went live and was ready to be extended to other work cells. |

Total investment for the Andon system was $1194 (implementing a single Tower Light), but this is easily extendable by acquiring new Tower Lights (each $315) [49]:

- TL70 Modular Tower Light ($315);
- DXM700 Wireless Controller ($579);
- Banner Cloud Data Services (CDS)—Starter ($300—Annually).

Quantitative Results

The Andon system was installed in January 2020 in the Munich manufacturing site of a biomedical company, as shown in Figure 7. The success of the solution was evaluated using two KPIs for six months, as shown in Table 4.

- The number of non-conformances measures the amount of non-conformances created in the production line and incoming inspection areas during a month. The number of non-conformances decreased by 63% since the introduction of the Andon system.
- The processing of non-conformance takes on average 10 working hours (root-cause analysis, definition of countermeasures, implementation of countermeasures, etc.). Savings are measured in working hours based on the jumping-off point (JOP), which was February 2020.

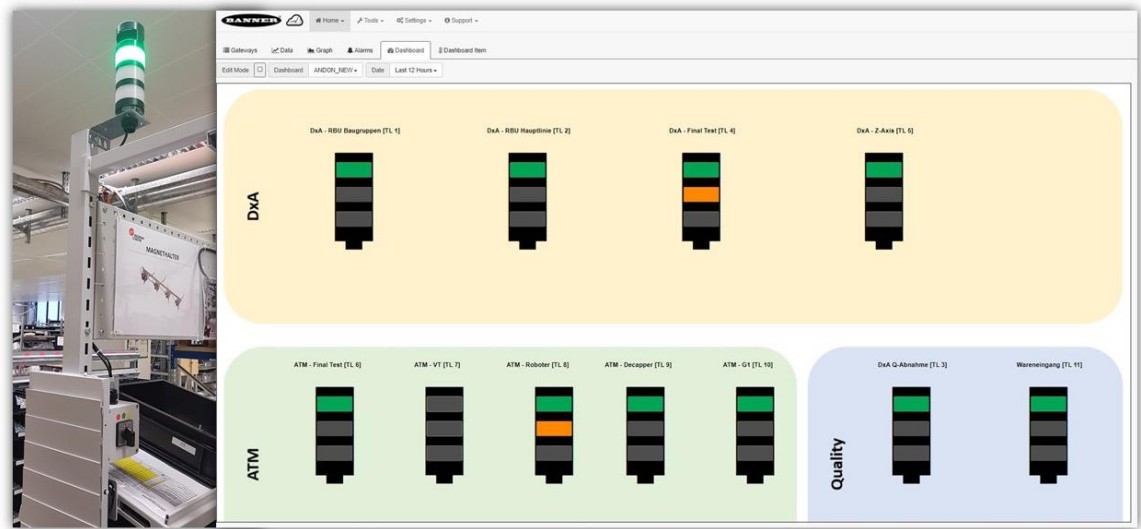

**Figure 7.** Implementation of the Andon system.

**Table 4.** Andon system—quantitative results. JOP: jumping-off point.

| Month/2020 | Savings to JOP | Number of Non-Conformances |
|:---:|:---:|:---:|
| February | - | 145 |
| March | 270 h | 118 |
| April | 650 h | 80 |
| May | 720 h | 73 |
| June | 850 h | 60 |
| July | 920 h | 53 |

### 3.3. Mixed Reality for Manufacturing and Shipping Processes

HoloLens coupled with the software solution Microsoft 365 Guides is a hands-free computer for the operator that achieves the following:

- It displays the manufacturing instructions as interactive Standard Operating Procedures (SOPs) and renders high-definition holograms that respond in a similar manner to physical objects. It goes where you go, sees what you see and does what you say;

- It displays digital pick lists. Order details are automatically assigned and sent to each picker, where they can follow instructions on their HoloLens, resulting in a more efficient process. A digital picking list can more easily include additional information such as product images, serving as an additional quality control check and as a validation that the picker is grabbing the correct item.

Mixed reality uses one of the technologies selected in Section 2.1.2: VR. The implementation following the roadmap is described in Table 5.

**Table 5.** Virtual reality—roadmap application.

| Step Number | Step Description | Application for This Solution |
|:---:|:---:|:---|
| 0 | Identify bottlenecks | Work orders for shipping are often picked incorrectly; paper-based pick lists are lost or are not properly reported to the database. |
| 1 | Develop a strategy | Two shipping orders per day, consisting of 10/20 items, need to be prepared and commissioned. One pair of mixed reality glasses that can be shared between the operators is needed. Additionally, the pick lists must be digitized. |
| 2 | Ideas and Prototypes | Standard shipping is used for the prototype, which consists of 15 items (10 crates and 5 boxes). The pick list needs to include customer information, the date and time of the order, the order number, the product location in the warehouse, a product locator and a photograph of the product. |
| 3 | Connect/plug-in your devices | One HoloLens is ready to be used and the digital pick list is accessible. One operator is trained on the usage of the mixed reality glasses, and they will perform a couple of dry runs. |
| 4 | Analyze | Control the picking time: it must be shorter. Correct possible mistakes: missing items in the picking list, ergonomics not properly controlled, etc. |
| 5 | Go Live | Once the final quantitative results are available, the tool is ready to go live and to be extended to all shipping orders. |

The total investment for the mixed reality system is $3520 [50]:

- HoloLens 2 ($3455);
- Microsoft 365 Guides ($65 per user/month).

Quantitative Results

The results of mixed reality implementation are not available at this point in the research. The initial qualitative results gained from the operators in the Munich manufacturing site are presented in Section 4 below.

### 3.4. Pick-to-Light—Boosting Productivity in the Picking and Kitting Processes

Manual picking processes that are not properly carried out can lead to significant monetary losses. Operators need experience to know the exact location of the bins or storage racks for the different work orders in the production line.

The pick-to-light (PTL) system provides an extraordinary solution: the front of each bin or storage rack is equipped with a numerical display with buttons. These devices or nodes are controlled by software. This software lights up the display, prompting the operator to remove goods from the bin or storage rack, and indicates how many units of the article in question should be taken on the screen.

Once the operator has finished extracting the corresponding units, they press a key to confirm that the operation has been carried out. This informs the software, and the light is turned off.

PTL Solution Kits consist of several PTL devices connected to a DXM700 Controller, which is configured via a touch screen.

The PTL uses two of the technologies selected in Section 2.1.2: the IoT and Cloud Computing. The implementation following the roadmap is described in Table 6.

**Table 6.** Pick-to-light—implementation process.

| Step Number | Step Description | Application for This Solution |
|:-:|:-:|:--|
| 0 | Identify bottlenecks | Work orders are often picked incorrectly, working processes are inefficient, the picking time is greatly increasingly and the increasing number of materials in the warehouse has visibility worsened. |
| 1 | Develop a strategy | Based on the number and complexity of the work orders to be picked, 10 lights and a controller are needed to improve the picking process in the warehouse. |
| 2 | Ideas and Prototypes | The most-picked work orders are selected. |
| 3 | Connect/plug-in devices | Pick-to-lights were installed, connected to the controller, a dashboard was set, and the warehouseman was made ready to analyze the performance of the picking process based on the lights. Some dry runs were performed until the system ran stably. |
| 4 | Analyze | The KPI was set to control the evolution of the picking time and analyze the results, asking the question "Am I winning or am I losing?". |
| 5 | Go Live | Once the final quantitative results are available, the tool is ready to go live and to be extended to other warehouse areas. |

The total investment for the PTL system is $2648 (for one PTL), and it is easily extendable by purchasing additional PTL devices [51]:

- PTL110S-TD3-QP150 ($149);
- SOLUTIONSKIT-PTL ($2499).

Quantitative Results

PTL results are not available at this point in the research. The initial qualitative results gained from the operators in the Munich manufacturing plant are presented in Section 4.

## 4. Discussion

### 4.1. Roadmap Comparison

As described earlier in Section 2.2, the proposed roadmap aims to simplify access to Industry 4.0 technologies for SMEs. The main differences between the roadmap proposed within this paper and roadmaps presented in previous research works are its focus on SMEs and the practical viewpoint. To highlight this extended approach, the following three criteria will be used in a weighted decision matrix, as shown in Table 7:

- Task-oriented: SMEs do not have the same structures as large enterprises and, therefore, the roadmap must be task-oriented rather than team-oriented. A roadmap that focuses on the completion of particular tasks makes it possible to create a roadmap with a single headcount, without involving several departments. This reduces the resources required to create the roadmap.
- Investment considerations: the roadmap will take into account the monetary limitations of SMEs for implementation.

- Practical implementation: the roadmap should not conclude with a theoretical development of the strategy; instead, it should continue with its deployment in practice and the subsequent measurement of the results through KPIs.

**Table 7.** Roadmap comparison.

| Proposed Roadmap | Task-Oriented | Practical Implementation | Investment Considerations |
|:---:|:---:|:---:|:---:|
| [21] | ● | ● | ● |
| [13] | ● | ● | ● |
| [17] | ● | ● | ● |
| This paper, Section 2.2 | ● | ● | ● |

The fulfillment of the abovementioned criteria is represented in Table 7 (green circle: fulfilled, red circle: not fulfilled).

*4.2. Data Obtained during Usage of the Roadmap*

Based on the quantitative results presented in Section 3.2 and on the qualitative results gained during the prototype phase, the Andon system improves the operation as follows:

- The key benefit of the Andon system is that it makes the condition of the manufacturing processes readily and easily visible to plant managers, operators and maintenance personnel, improving cross-functional collaboration;
- In addition, it makes it far easier to ensure that processes are being carried out efficiently and productively;
- Andon boards can also act as an early warning device, improving visibility. When abnormal conditions are noted, they will appear on the Andon board, giving operators and managers time to correct problems before they begin in some cases.

The qualitative results gathered during the prototype phase show that the implementation of a PTL system reaps great benefits:

- It makes the operations far more flexible by decreasing the number of operator movements. In addition, it takes advantage of the time previously spent on reading, writing down and checking information to prepare more orders, thereby increasing employee productivity;
- It drastically decreases errors made in order preparation;
- Assistive picking technology requires the least training for new hires. This facilitates the onboarding of operators in facilities;
- It consists of a mature technology that has incorporated increasing possibilities regarding the design of the devices and the nature of the messages displayed. This, consequently, provides enhanced customization options.

The qualitative results gained during the prototype phase show that the usage of mixed reality glasses promotes the following benefits:

- It reduces the gap between inexperienced employees and veterans in many professions, due to the fast learning curve;
- It is not a printout or a static document—there is real-time interaction and the picking list can be updated in real-time;
- It simplifies complex problems/situations through the usage of high-definition holograms, increasing the productivity of operators.

## 5. Conclusions

The key findings of this research are as follows:

- The roadmap helps SMEs to enter the new era of manufacturing by using a simple process that does not require high-level expertise or large teams. The proposed roadmap covers the first gap highlighted in Section 1;
- The three Industry 4.0 technologies selected (Cloud Computing, IoT and VR) are available for every company, no matter the number of employees or its revenue. According to the second gap identified in Section 1, this research has shown the implementation of Industry 4.0 technologies from a practical point of view using the roadmap;
- The initial investment is affordable for every SME (less than €4000 in the three results presented) and it is within the range presented in Section 2.1.4; the solutions have evolved into low-cost products that can generate great productivity improvements.

Moreover, the key characteristics of the SAL presented in Section 2.3 are visible in the three technologies presented during this paper, as shown in Table 8. Further analysis of the quantitative results for the three tools is still required.

**Table 8.** The Smart Assembly Line—expectations fulfilled. PTL: pick-to-light.

| Tools | Collaboration | Data-Driven | Preventive Maintenance | Proactive Assembly Line |
|---|---|---|---|---|
| Andon System | ✓ | ✓ | ✓ | ✓ |
| Mixed reality | ✓ | ✓ | ✗ | ✓ |
| PTL | ✓ | ✓ | ✗ | ✓ |

In addition, the Industry 4.0 portfolio is very large, and the lack of standardization still makes it difficult for SMEs to choose the right IoT technology. The next step in this research would be to create a web-based application based on the roadmap that guides users in SMEs through the different technologies available. The application will require an input—e.g., a questionnaire—and finally propose the best possible solutions based on the requirements and budget available.

**Author Contributions:** A.C.: conceptualization, methodology, investigation, writing—original draft preparation, and visualization; M.A.S.: conceptualization, methodology, investigation, writing—review and supervision; C.G.-G.: conceptualization, methodology, investigation, writing—review and supervision. All authors have read and agreed to the published version of the manuscript.

**Funding:** This research was funded by the Annual Grants Call of the International Doctorate School of the Spanish National Distance-Learning University (EIDUNED).

**Acknowledgments:** This paper has been produced within the scope of the doctoral activities carried out by the lead author at the International Doctorate School of the Spanish National Distance-Learning University (EIDUNED_ Escuela Internacional de Doctorado de la Universidad Nacional de Educación a Distancia). The authors are grateful for the support provided by this institution.

**Conflicts of Interest:** The authors declare no conflict of interest.

## Abbreviations

The following abbreviations are used in this manuscript:

| | |
|---|---|
| DT | Digital Transformation |
| IfM | Institute for Manufacturing |
| IoT | Internet of Things |
| JOP | Jumping-off point |
| KPI | Key Performance Indicator |
| PTL | Pick-to-light |
| SAL | Smart Assembly Line |
| SME | Small and Medium-Sized Enterprises |
| SOP | Standard Operating Procedure |
| VR | Virtual Reality |

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
