# Peer review of "Industry 4.0 Roadmap: Implementation for Small and Medium-Sized Enterprises"

_applsci, doi:10.3390/app10238566_

Round 1

Reviewer 1 Report

The paper focuses on investigating from a practical point of view Industry 4.0 roadmap for small and medium-sized enterprises. It is not clear what the research methodology is adopted. It is also not clear how the roadmap formed and whether it is of research value. In addition, it is not clear how to evaluate the result (e.g. roadmap) and justify its research contribution.

  • In the abstract, please highlight the main ideas of roadmap and the result.
  • In section 2, please follow the flow of three phases as indicated in the beginning of the section.
  • The literature review is quite thin and trivial. The authors need to summarize the main points for those papers concerning theoretical viewpoint and why there fail to address the research issue from practical viewpoint. There are rich literatures on the related areas, such as IT adoption, strategy of digital transformation, etc.
  • There are several “Error! Reference source not found.”. Please fix them.
  • Please explain how to get Figure 1 and please provide the reference.
  • Page 4. “But there are two technologies that are the pillar to begin the transition to the 4th industrial revolution and that has been analysed during this research: the digital transformation and the Internet of Things.” The digital transformation and the Internet of Things seem on different levels. It is required to explain what exactly the paper focuses on.
  • Page 5. It is not clear what the main points are in section 2.5 for the whole paper.
  • Page 5. “Based on the systematic review, benefits of Industry 4.0 are well identified.” It is not clear systematic review is referring and what benefits are identified.
  • Page 5. It is not clear how to come out the roadmap. Is there anything new in the roadmap?
  • Page 6. Please check the reference [0].
  • Page 6. Is there any reference for the definition and four key characteristics of Smart Assembly Line?
  • Page 6. “shall cost less than $5000”. How to justify the amount.

Reviewer 2 Report

The proposed paper introduces a six-step Roadmap Industry 4.0 applicable for small and medium-sized enterprises.

The abstract should state briefly the purpose of the research, the principal results and major conclusions. An abstract is often presented separately from the article, so it must be able to stand alone. In my opinion, the current version does not encourage the reader to read the article.

The authors have proposed their solution. Unfortunately, the literature review is very short and does not include a critical review of the present literature on SM/Industry 4.0.

The authors introduce figures and tables but do not refer to them in the text. Additionally, there are editing errors in the article, e.g. lines 101, 109, 130, 211, 215, 216..... ; reference [0], etc.

Reviewer 3 Report

  1. Detailed English proofreading is required. (e.g. Employees will not be able to return to their workplaces as it used to be before: less occupancy 143 per room, social distancing, …)
  2. Title and abstract deliver just a general concept. They need to be changed for more clear unique opinions for SME’s industry 4.0 introduction.
  3. In Section 1, more clearly define DT, as has multiple definiens and applications.
  4. This paper works following three phases- literature reviews, introduction to the roadmap and verification. However, the reviewed literatures are very short. More Intensive reviews and their comparisons are required. Then, they compared with the proposed roadmap. Without, the roadmap can be evaluated as an efficient roadmap for SME.
  5. Moreover, the testing and validation have to be proven more quantitatively.
  6. Check a reference error in Section 2.4.
  7. The roadmap shown in Figure 4 is too simple to capture the overall 4IR technologies for SMEs.
  8. The title has “Smart assembly line (SAL)”. However, the explanations provided in Section 2.7. are too general and simple.
  9. Section 3 must be redesigned and rewritten. The detailed implementations and proofs are required. Currently, it is difficult to capture the real implemented SME’s example.
  10. Similarly, the discussion and the conclusion sections have to be rewritten. These are provided without any logical evidence.
  11. The provided framework lacks uniqueness and advantages. Examine other similar frameworks for SME and compare it.

Round 2

Reviewer 1 Report

The revision is satisfactory. 

Reviewer 3 Report

  1. Several reviewer’s comments are explained in the revised version. In particular, the literature review sections are improved well.
  2. However, the critical issue is not explained. The provided implementation (including figure 7) doesn’t answer why it is the practical roadmap or methodology. As the manuscript aims to industry 4.0, factory-level or production-level implementation have to be provided and compared with other methodologies. Still, it lacks the evidence of the provided concept.
  3. The changed title “practical industry 4.0 roadmap for small and medium-sized enterprises” gives ambiguous to readers. The current manuscript gives a concept for industry 4.0 roadmap.
  4. Point 7 (explained by authors) doesn’t answer enough 4IR technologies for SMEs. Still, it is too simple.
